# Ripplocations provide a new mechanism for the deformation of phyllosilicates in the lithosphere

Joe Aslin [1], Elisabetta Mariani [1], Karl Dawson[1] & Michel W. Barsoum [2]

Deformation in Earth's lithosphere is localised in narrow, high-strain zones. Phyllosilicates, strongly anisotropic layered minerals, are abundant in these rocks, where they accommodate much of the strain and play a significant role in inhibiting or triggering earthquakes. Until now it was understood that phyllosilicates could deform only by dislocation glide along layers and could not accommodate large strains without cracking and dilation. Here we show that a new class of atomic-scale defects, known as ripplocations, explain the development of layer-normal strain without brittle damage. We use high-resolution transmission electron microscopy (TEM) to resolve nano-scale bending characteristic of ripplocations in the phyllosilicate mineral biotite. We demonstrate that conjugate delamination arrays are the result of elastic strain energy release due to the accumulation of layer-normal strain in ripplocations. This work provides the missing mechanism necessary to understand phyllosilicate deformation, with important rheological implications for phyllosilicate bearing seismogenic faults and subduction zones.

---

[1] University of Liverpool, Liverpool L69 3BX, UK. [2] Drexel University, Philadelphia, PA 19104, USA. Correspondence and requests for materials should be addressed to J.A. (email: joeaslin@liverpool.ac.uk)

Layered phyllosilicate minerals (or sheet silicates) include clays, serpentine and micas, and are important constituents of Earth's lithosphere, occurring in a range of rock types and tectonic settings. Laboratory studies show that interconnected layers of phyllosilicates are weak[1,2], while in the field these minerals are observed to be ubiquitous in plate bounding faults and shear zones[3,4]. Due to their weakness, phyllosilicates are recognised to localise deformation and their rheology has a strong influence on the nucleation and propagation of earthquakes in seismogenic faults[5–7] and on the dynamics of subduction zones and zones of shear in the viscous lower crust[3,8]. The mechanical response of phyllosilicates is due primarily to the strongly layered, and therefore anisotropic structure of their crystal lattice. This consists of layers of silicon-oxygen tetrahedra (T) and octahedrally coordinated cations (M) referred to as TMT layers (Fig. 1). In micas and most clays these layers are bound together by charge-balancing interlayer cations (such as K in biotite; Fig. 1a) and possess an overall negative charge. In serpentines and most chlorites, tetrahedral-octahedral layers are linked by Van der Waals forces and hydrogen bonding and their charge is neutral (Fig. 1b, c). The layering described makes phyllosilicates particularly weak when sheared parallel to layers (the basal or c planes)[9] and gives them a disproportionately large influence on the strength of the rocks in which they occurr[3,10–14].

Currently, dislocation creep, a viscous deformation mechanism, is understood to occur in minerals by the motion of defects called dislocations. This process is a combination of dislocation glide, where the defects move along specific layers and directions in the crystal lattice (known as slip systems) and dislocation climb, or cross-slip whereby dislocations are able to step between planes in a crystal lattice to avoid obstacles such as impurities. The strongly layered structure of phyllosilicates, however, limits dislocation glide to the basal plane with no facility for dislocation motion on other planes[9]. As a result, dislocation creep is not a viable deformation mechanism in phyllosilicates and the means by which strain is accommodated parallel to the c-axis (perpendicular to the layering) remain ambiguous. In addition to this, the work of Noe and Veblen[15] questions the viability of dislocation defects within the (001) biotite interlayer (the cleavage plane) altogether, on the basis of energy considerations. These authors show that instead, basal dislocations may be found only within the (001) oxygen layer between the octahedral and tetrahedral sheets, thus highlighting that it is a misconception to assume that cleavage planes are glide planes for dislocations.

In layered minerals kinking is observed to be the dominant process of shortening parallel to basal planes (Supplementary Figs. 1 and 2). However, kink bands (KBs) cannot be explained using glide of basal dislocations alone. The most comprehensive models invoke the formation of complex arrays of dislocation walls at KB boundaries (KBBs), which impart curvature of the lattice over a finite region[16,17]. While these models determine basal slip as the primary mechanism for kinking, they recognise that another mechanism is necessary to account for the c-axis parallel strain required to form KBs[16] (Supplementary Note 1).

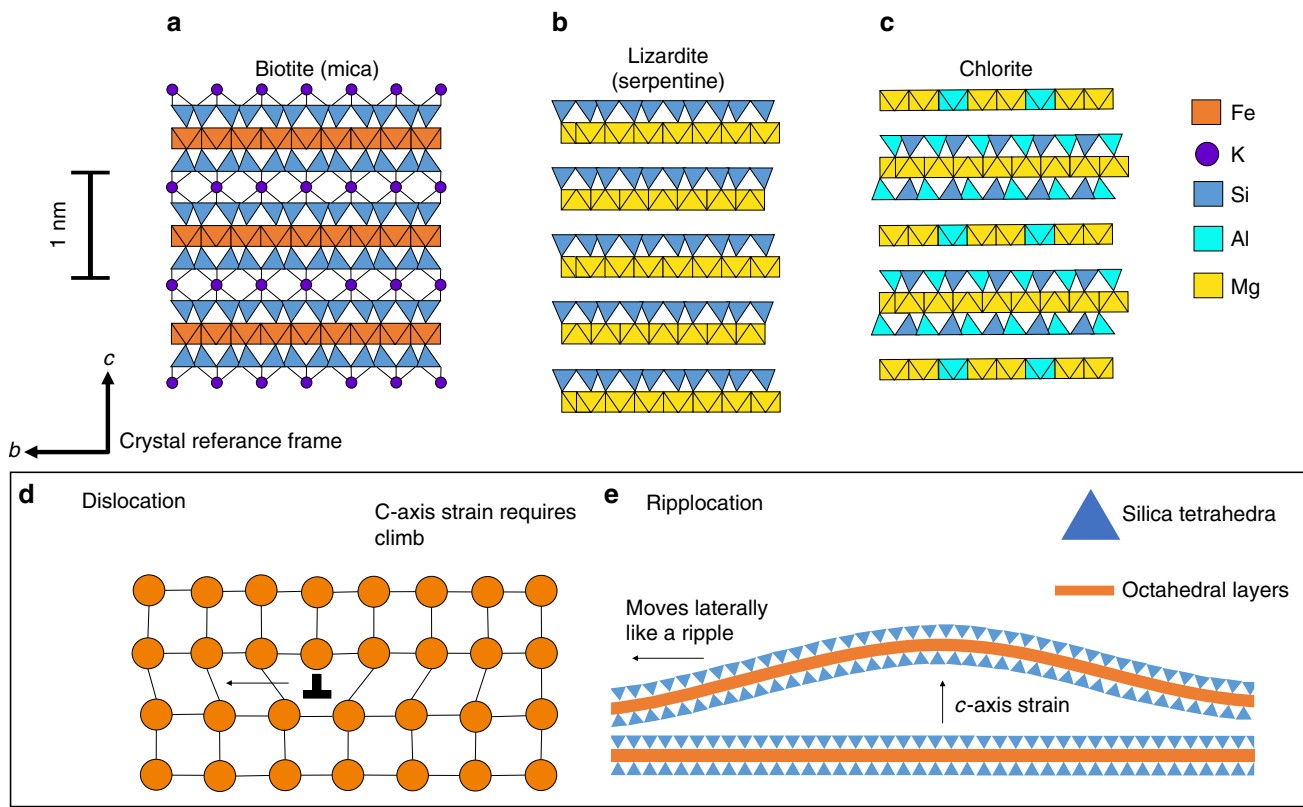

**Fig. 1** Phyllosilicate structures and ripplocation morphology. **a–c** Atomic structure models of some common phyllosilicates with valences identified by polygons. The indices of these polygons are occupied by O atoms or OH groups. **a** Biotite, like other micas, requires an interlayer cation (I) to balance the overall negative charge of its tetrahedral (T) and octahedral (M) layers. In the cases of biotite and muscovite this cation is predominantly K, whereas in paragonite, and others, it is Na. **b** Lizardite, like the other serpentine minerals antigorite and chrysotile, has a 1:1 ratio of T to M layers, which possess a neutral charge resulting in no interlayer cations. **c** The structure of chlorite is similar to biotite except that the interlayer cations are replaced by individual octahedral layers, so called brucite like sheets. **d**, **e** Comparison between a dislocation (**d**) and a ripplocation in a generic phyllosilicate lattice (**e**). Both enable the movement of one plane of atoms over another. However, while dislocation climb is required to deform out of this plane in **d**, ripplocations intrinsically contain an element of c-axis parallel strain due to bending and more crucially are attracted to each other rather than repelled

To date this *c*-axis strain has been explained through the brittle damage mechanisms of dilation or cleavage when basal glide is inhibited[17,18]. The importance of phyllosilicates with regard to the strength and dynamics of the lithosphere demands a proper understanding of their behaviour. Despite this, existing mechanisms fail to fully explain commonly observed structures and processes associated with phyllosilicate deformation, suggesting an important piece of the puzzle is missing.

Recently, a new fundamental defect type, named ripplocations, has been proposed, which takes the form of atomic-scale ripples in the basal layer[19]. They have so far been studied and modelled in $MoS_2$[19], $Ti_3SiC_2$[20,21] and graphite[21,22] but are theoretically applicable to phyllosilicates and other layered solids[23]. Ripplocations have a similar effect to dislocations in that they allow the motion of one plane of atoms over another and may be more favourable in certain crystallographic directions[19,24], but differ because they do not require the breaking and re-building of in-plane bonds (Fig. 1d, e) and result in an expanded unit cell dimension normal to the weak layer. Another fundamental difference is that ripplocations attract each other and may merge, in contrast to same-sign dislocations, which repel and pile-up[19,21]. Ripplocations on different layers are also capable of aligning to form a through-going bend in the lattice, i.e. a KBB[21,23]. Crucially, ripplocations involve a ripple in the basal layers, giving them a component of *c*-axis parallel strain that is absent in basal dislocations. They have been visualised in transmission electron microscope (TEM) observations of experimentally deformed $MoS_2$[19] and $Ti_3SiC_2$[20,21], while ripplocation behaviour has been tested through numerical modelling[19,21]. However, prior to this study, ripplocations have not been reported in any naturally occurring mineral. The geological implications of the presence of ripplocations in phyllosilicates are substantial.

They provide a previously unrecognised mechanism to account for the geometric problems associated with deforming such anisotropic minerals. Phyllosilicates are increasingly seen as fundamental players in the localisation of strain in the lithosphere, with a controlling effect on the rheology of many crustal rocks[3,7,8]. Our current understanding of the deformation of these important minerals is hindered by the lack of a mechanism for geometrically necessary *c*-axis strain. Ripplocation motion and their interactions may be that missing mechanism. In this study we use high-resolution TEM to examine the nano-scale structure of naturally deformed biotite mica from a regional scale shear zone in the Ivrea-Verbano area of North-West Italy (Supplementary Note 2), and show that ripplocations are essential to account for the lattice bending and basal delaminations we observe.

## Results

**TEM observations of ripplocations in biotite**. TEM samples were prepared using the focussed ion beam (FIB) in situ lift out technique[25] (see Methods). All biotite grains analysed contained regions of lattice expansion parallel to the basal plane, these were up to 0.2 μm in length and visible as bright streaks in brightfield images (Fig. 2, Supplementary Fig. 3). The width of biotite interlayers expands from 3 Å in the undisturbed lattice to up to 60 Å in the regions of greatest lattice expansion. The TMT layers by contrast remain at 7 Å width but form ripples deflecting around regions of lattice expansion (Fig. 3). Expanded regions, transitioning to delaminations in the brightest areas, are often asymmetrical, lozenge-shaped and occur in en-echelon stacks, which are aligned in two principal directions, 30°–60° apart and bisected by the basal plane (Fig. 2a–d). This geometry produces an elongated diamond-shaped pattern across entire biotite grains (Fig. 2a, Supplementary Fig. 4). Expansion structures have the

appearance of waves (Fig. 3) and exist at multiple scales, from delamination of two TMT layers, along the interlayer, visible at the micron-scale, to ripples that increase interlayer distances at the nano-scale (arrows in Fig. 3a). Such large rotations of layers forming ripples are orders of magnitude greater than those which could be produced by individual dislocations or complex arrays of dislocation walls, which can result in bending of the (001) plane only within a broader finite region on the order of 1–2 μm[16,17]. Bending of (001) within individual KBBs has been described on the scale of 40° in 0.2 μm[17] and was interpreted to have been achieved by the rotation accumulated across numerous dislocation walls. However, we observe rotation of very low numbers of (or in some cases individual) biotite TMT layers on the order of 30° across just 3 or 4 nm (Fig. 3). According to the above studies, individual dislocation walls are capable of rotating (001) by up to 2°, this would mean our observations require around 15 individual dislocation walls within the space of 3 or 4 nm, a distance of <8 unit cells in the *a* direction and around 4 unit cells in the *b* direction of biotite. This degree of rotation cannot be physically accommodated by means of dislocation walls but can be explained by elastic bending of basal layers as in existing models of ripplocations[19,21]. Profiles drawn across expansion structures concur that regular 10 Å spacings (the *c*-axis dimension of a biotite unit cell) increase, and that this expansion occurs within the interlayer regions (Fig. 3a–c).

Most previous studies on interlayer delaminations in phyllosilicates did not utilise the FIB sample preparation method but rather used ion milling systems, which involve polishing until a perforation is produced in the centre of the sample (see Methods). Therefore, for direct comparison with previous work, and to eliminate the FIB method as a potential source of delamination structures, we also prepared foils from three samples using an Ar precision ion polishing system (PIPS). At the ultrathin edges of the central perforation of PIPS specimens, voids could occasionally be seen to grow during observation. They often became much wider than delaminations in FIB specimens or within the thicker regions of PIPS samples and had lower aspect ratios. In contrast, thicker regions further away from the perforation edge contained smaller expansion structures and delaminations closely resembling those of the FIB specimens in size, aspect ratio and their arrangement in diamond-like arrays (Supplementary Fig. 5). To examine the relationship between these arrays and the degree of strain a biotite grain has experienced, samples were prepared (using both FIB and PIPS methods) from undeformed Westerly granite (WG) specimens to be compared directly to those from mylonitic biotite described above and shown in Figs. 2 and 3 (Supplementary Table 1). Diamond-shaped arrays of expansion structures and delaminations were present in all the biotite samples analysed (Supplementary Figs. 4–6) but were more clearly observed and more abundant in biotite from naturally deformed mylonite samples. Evidence for ripplocations in nominally undeformed granitic biotite is not entirely surprising and might be linked to two possible causes: (1) ripplocations are a form of crystal defect and might therefore develop during crystal growth, due for example to layer mismatch, much like the way growth-related lattice mismatch forms dislocations[26]; (2) an undeformed granite, during intrusion through the Earth's crust, could develop local grain-scale differential stresses due to crystal impingement during growth or differential thermal expansion/contraction.

It must be noted that we did not observe kinking in biotite grains from WG, suggesting that while ripplocations are present in this biotite, no motion or interaction of ripplocations is likely to have occurred. This observation may support the idea of ripplocations as growth defects (point 1). The evident increased abundance of lattice expansion and delaminations in biotite from

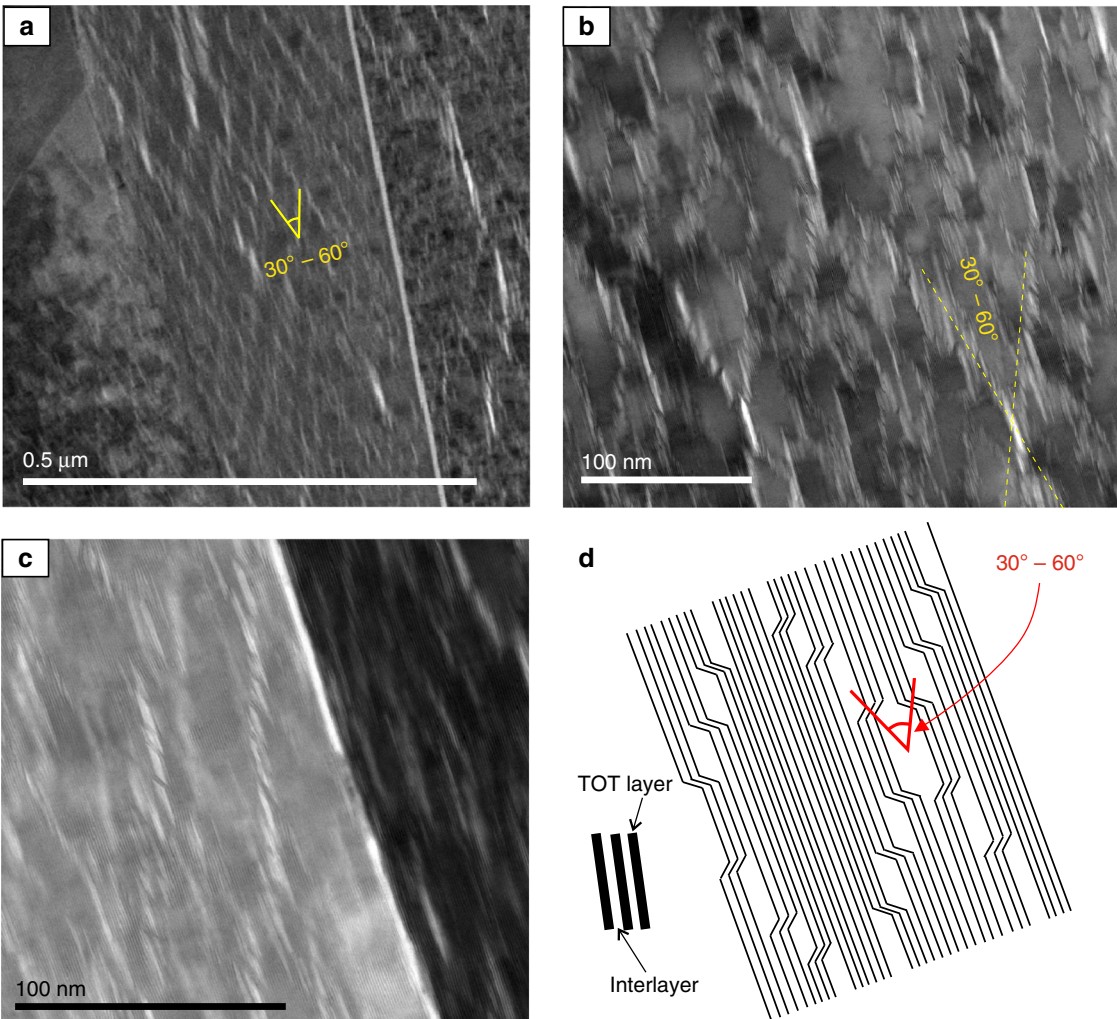

**Fig. 2** Diamond-shaped expansion structure arrays. Brightfield transmission electron microscope micrographs of focussed ion beam-prepared biotite specimens from mylonitic orthogneiss. **a** Bright streaks parallel to the basal plane, but stacked in two principle orientations, produce elongated, diamond-shaped arrays across the entirety of biotite grains. The acute angles of these diamonds measure between 30° and 60°. **b** At higher magnifications, these streaks are revealed to be up to 60 Å wide, lozenge-shaped lattice expansion (or delamination) structures of the biotite basal planes. **c** Detail of expansion structures/delaminations stacked in en-echelon arrays. **d** Schematic of the features observed in **a**–**c** showing the en-echelon stacking of lozenge-shaped expansion structures/delaminations offsetting TMT layers and producing diamond-shaped arrays

mylonites is indicative of the intense deformation experienced by these rocks relative to granites.

The best explanation for the geometric arrangement of conjugate lattice expansion arrays we show in this study is an overall expansion parallel to the $c$-axis (perpendicular to the basal plane) of the biotite grain. The fact that the edges of these 'diamond' geometries are 30°–60° apart and are bisected by the basal plane is not a coincidence, it is a clue to the stress field in which these structures formed. The clear conceptual comparison is to conjugate normal faulting, which occurs under extensional stresses, where $\sigma_1$ is vertical and $\sigma_3$ is horizontal and parallel to extension (Fig. 4a). The acute angle between faults is commonly 60° but can vary and the effect on the rock mass is an extension perpendicular to the plane that bisects this angle[27]. In our diamond-shaped arrays, the bisecting plane is the basal plane and the extension (and $\sigma_3$) would therefore be parallel to the $c$-axis (Fig. 4b). These conjugate arrays are therefore likely to have formed during expansion parallel to the $c$-axis (normal to the basal planes).

Since, in order to achieve electron transparency, TEM specimens are exceptionally thin (<100 nm), the possibility arises that

the delaminations form to relieve stored $c$-axis strain once the grain becomes completely unconfined during thinning[23]. Basal dislocations are not a viable mechanism for storing this strain energy as they possess no component of $c$-axis strain. We propose that this $c$-axis strain could initially be stored in the form of bulk ripplocations, which originate from compression parallel to the basal planes (as a component of the overall stress field deep in Earth's lithosphere) and their strain energy is released to form conjugate arrays of expansion structures on thinning for TEM analysis. This interpretation is supported by substantial flexing of the TEM films observed during FIB preparation (Supplementary Fig. 7), indicating elastic energy release.

Delaminations have been previously observed in TEM studies of deformed micas[28–31] and attributed to electron beam damage[28]. However, the extensive, ordered diamond-shaped arrays described here have not been reported before. To assess whether the structures seen are caused, or extenuated by beam damage, we analysed regions previously unexposed to the electron beam. The first evidence that abundant delaminations are not beam damage artefacts is the fact that they are visible at the outset of observations in the TEM and cover grains

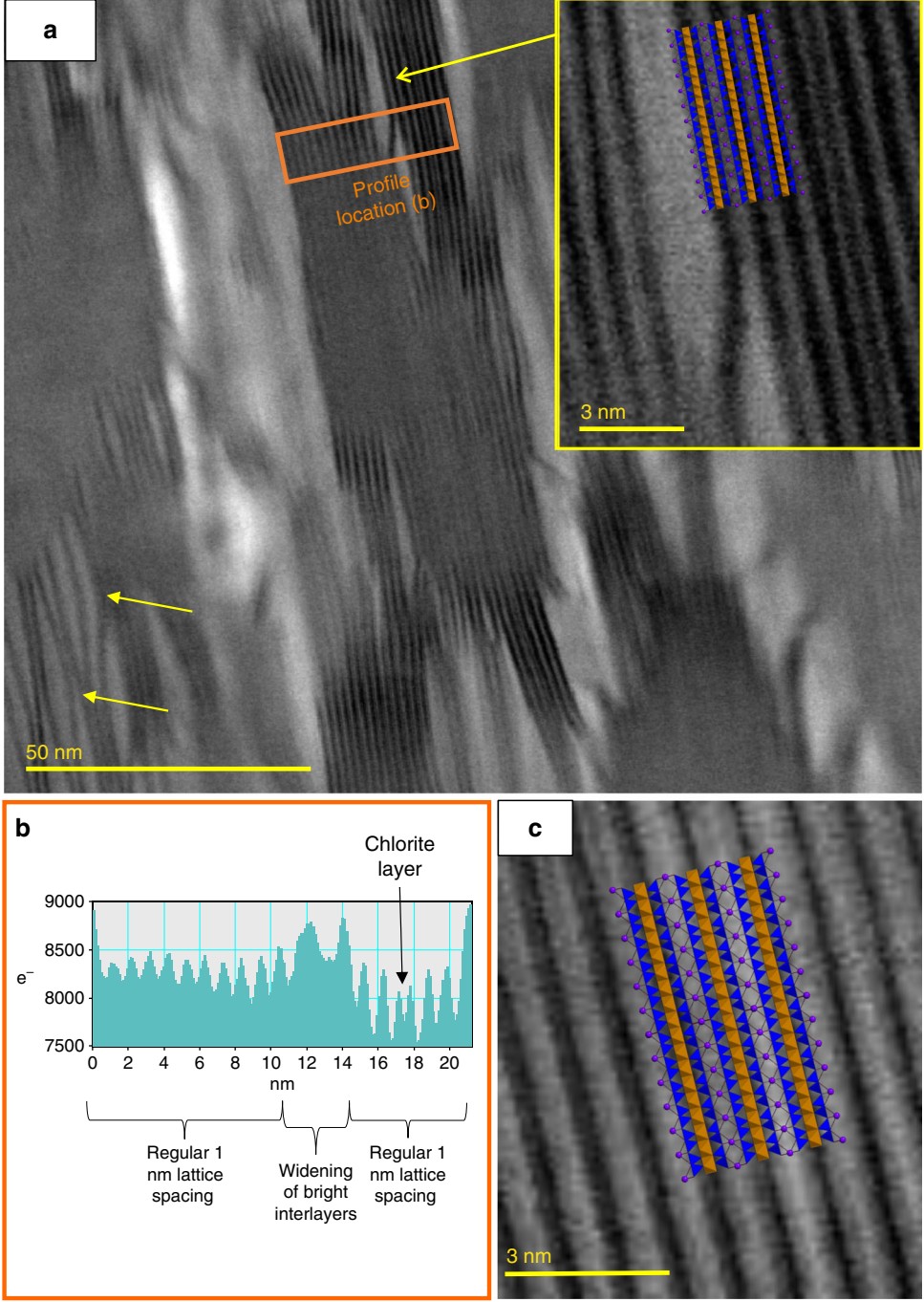

**Fig. 3** Lattice curvature and interlayer expansion. **a** Lattice curvature is visible at the scale of individual TMT layers, which bend up to 30° within as little as 3 nm forming ripples (indicated by arrows in bottom left). Inset shows detail of the region indicated by the attached arrow overlain with a biotite atomic structure model showing how the lattice expansion occurs within the interlayer. **b** Profile integrated from the region defined in **a** showing a widening of brighter contrast interlayers within the regular 1 nm lattice spacing of pristine biotite. The double peak to the right represents an individual chlorite layer. **c** High-magnification transmission electron microscope lattice fringe image with biotite atomic structure model overlain. Dark regions centre on octahedral layers, intermediate (grey) regions show tetrahedral layers and light regions demonstrate the location of interlayers

extensively and uniformly (Supplementary Fig. 8a). Beam effects were observed during our experiments, but took the form of progressive mottling and amorphisation, eventually obscuring or destroying the lattice structure (Supplementary Fig. 8a, b). Beam damage effects are expected and observed to be localised to areas where the beam has been held and should not occur uniformly across a whole grain.

The principal mechanism previously provided for the formation of basal delaminations is the diffusion of alkali interlayer cations (Na in paragonite) due to beam exposure[28], with layer separations forming to counteract the resulting volume decrease. This was supported by a rapid decrease in the Na peak in sequential X-ray spectra of paragonite collected during progressive beam exposure. Interactions between alkali elements and an electron beam result in a depletion of the element's concentration at the sample surface and higher concentrations immediately below[32,33]. This must be accounted for during electron microprobe analysis of geological materials[34,35]. In our study, K is the

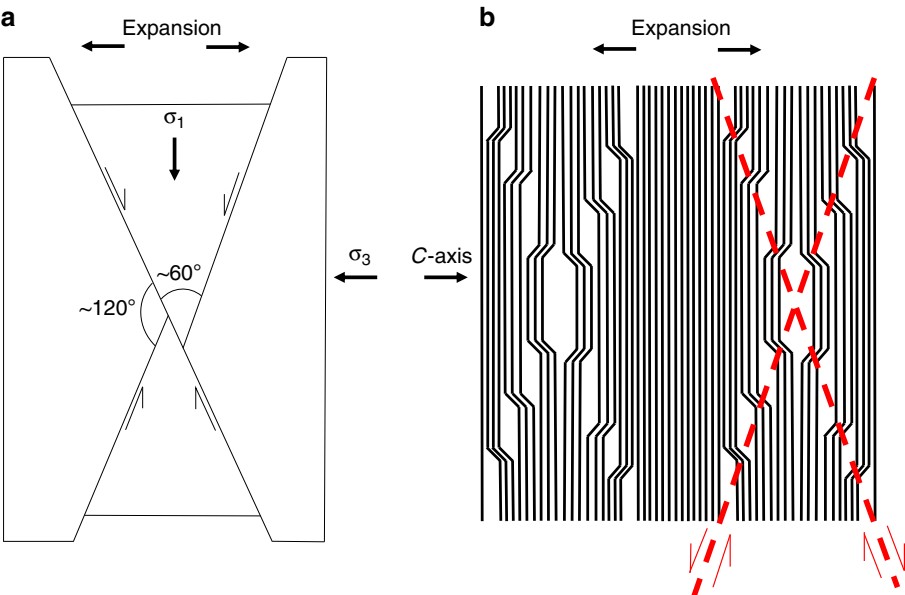

**Fig. 4** Formation of conjugate arrays of expansion structures. Analogy between **a** the Anderson model of conjugate faulting and **b** a schematic representation of the conjugate arrays of expansion structures and delaminations we observe in biotite. In both cases structures (faults in **a** expansion structures in **b**) form in two principal orientations, which are up to 60° apart and display opposite shear sense. The obtuse angle between the two orientations is bisected by $\sigma_3$ in **a** and the c-axis in **b**. In both cases this is the orientation along which expansion (or extension) occurs. In this way the diamond-shaped expansional arrays are crystallographically controlled as is evidenced by the fact that they have different orientations in different grains, being always aligned with the c-axis of each grain. Conjugate delamination arrays represent the most efficient way to relieve stored grain-scale c-axis strain energy, just as conjugate normal faults result from extension on a larger scale. The orientations of principal stresses in this figure are therefore representative of the time when stored elastic strain energy is released (during thinning for TEM), not necessarily the time during which strain was being accumulated (deformation within the viscous shear zone)

interlayer cation in biotite, it is heavier than Na and, under the same beam conditions, is also less mobile due to its lower diffusivity[32]. Despite this, the delamination arrays we document are more extensive than those reported in paragonite (Fig. 1a of Ahn et al.[28]) suggesting that diffusion of alkali cations may not be the cause. Energy dispersive X-ray (EDX) spectroscopy of our specimens in the TEM shows a decrease in K relative to Si over periods of up to 16 min, as could be expected from cation diffusion[35], however the rate of decrease slows over time and stabilises asymptotically around values of 0.33 K/Si (Supplementary Figs. 8c, 9–13). This is in contrast to the rapid and near-total loss of Na in the paragonite analysed by Ahn and co-workers[28]. In addition, in our observations K loss cannot be linked with a simultaneous nucleation or growth of delaminations because these are present before any K loss occurs. While the beam damage mechanism[28] does not therefore explain our structures, the delamination of basal planes must indeed have been driven by some degree of c-axis parallel strain. As basal dislocations cannot impart c-axis strain, we propose ripplocations as the defects governing the observed (001) interlayer plane bending and associated expansion.

## Discussion

Our model for the formation of conjugate expansional arrays is in agreement with numerical modelling of ripplocations in $Ti_3SiC_2$ and graphite[21], which suggests ripplocations, when confined, nucleate as multiple small ripples, but when unconfined, fewer, larger undulations are energetically favoured. This is analogous to the production of delaminations in our samples when the 'confinement' is removed during thinning. In other words, the energy of the ripplocations is unable to delaminate the layers around them until the confining energy is sufficiently reduced. Multiple small ripplocations build up whilst the rock containing the biotite

is experiencing differential stress in the Earth (Fig. 5a, b). If these stresses continue, or increase, then eventually the ripplocations gain enough energy to overcome the constraining force (lithostatic pressure) and may migrate, interact or accumulate across layers to form KBBs, hence the kinking of phyllosilicates in deformed rocks (Fig. 5c). If the ripplocations do not exceed the required energy to become mobile and form kinks, then they continue to store that strain energy until the confining energy is reduced, for example during TEM sample preparation. They then release c-axis strain energy with many small ripplocations on the same plane merging to produce fewer, larger expansion structures or delamination features, which form in conjugate extensional arrays (Fig. 5d). Through this model, ripplocations provide the c-axis driving force necessary to produce the basal delaminations observed in our samples and reported in previous studies. The model also provides the c-axis strain necessary to explain the pressure sensitivity measured in phyllosilicate-rich rocks (whereby their yield strength increases proportionally to the applied lithostatic pressure) up to their dehydration temperature[13]. In phyllosilicates both viscous (dislocation glide) and brittle (cracks associated with kinking and dilation) deformation mechanisms are seen to occur through a large range of conditions in the Earth (from the Earth's surface to the middle and lower crust[13,36–38]). In addition to this, while pressure sensitivity might traditionally be associated with brittle deformation, it occurs in micas even when no obvious microstructural evidence for classic brittle deformation can be observed[13]. Essentially, micas might deform by a pressure sensitive but non-brittle mechanism at a range of conditions. This has to date been a poorly understood phenomenon. Ripplocations resolve this conflict by providing a pressure sensitive mechanism of deformation that does not result in fracturing or other characteristic brittle microstructures (see also Supplementary Note 3).

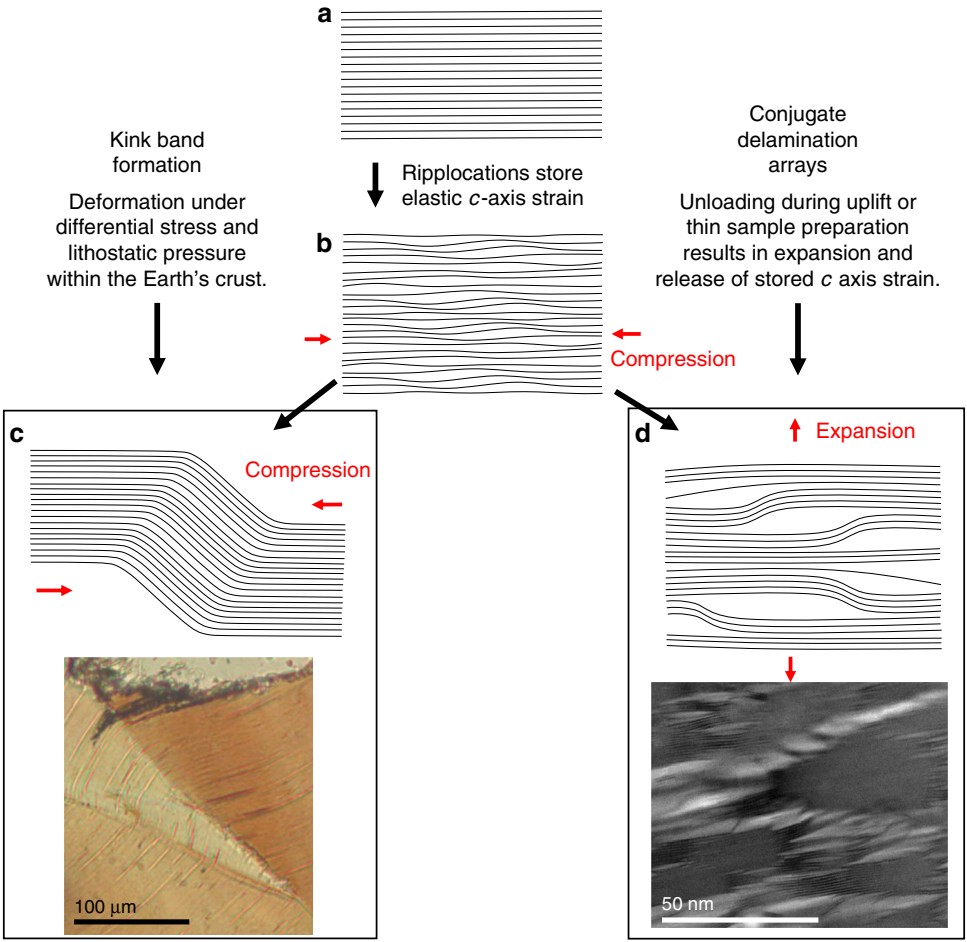

**Fig. 5** Conceptual model showing how ripplocations lead to kink bands or conjugate expansion structure arrays. A pristine undeformed biotite lattice (**a**) experiences differential stress with some component of compression parallel to the basal planes. As the lattice is confined this induces numerous small bulk ripplocations, which store the resulting $c$-axis parallel strain energy (**b**). Continued basal plane parallel compression imparts enough energy for ripplocations on different layers to migrate and merge to form a kink band between two kink band boundaries of opposite sign (**c**). The elastic $c$-axis strain energy becomes permanent strain on the formation of mature through-going kink bands. This may produce the kinked microstructures common in micas from viscously deformed rocks. Parts of a biotite lattice that have experienced a stress field similar to **b** but in which kinking has not been induced still contain stored elastic $c$-axis strain energy in the form of ripplocations. This energy is released on thinning for TEM analysis in the form of conjugate arrays of expansion structures and delaminations, which facilitate a grain-scale $c$-axis parallel expansion (**d**). This is represented by the structures shown in Fig. 2 of this study

In PIPS prepared samples, exposure to the beam did appear to initiate further distortion and growth of voids in ultrathin regions close to the perforation edge. It is possible that the small amount of energy imparted by the electron beam was enough to release further elastic strain from existing ripplocations in these regions, especially where a free sample edge could be exploited. This still requires the storage and release of some degree of $c$-axis strain, which cannot be explained by basal dislocations. The fact that these effects are only observed at ultrathin perforation edges and not in thicker regions or in FIB specimens, could explain why previous studies, which utilise PIPS-style methods, have reported large delaminations that grow under the beam[28] and have not reported conjugate expansion structure arrays.

This work demonstrates the existence of ripplocations within the phyllosilicate mineral biotite and highlights the fact that ripplocations may be a ubiquitous deformation mechanism in all types of phyllosilicates as they are applicable to all layered solids[23]. In particular, serpentine minerals and chlorite may be even more susceptible to the nucleation and motion of ripplocations than micas. Their structures do not incorporate interlayer cations and their interlayer bonding relies on weak Van der Waals forces and hydrogen bonds[39] (Fig. 1b, c). In clay minerals the impact of ripplocations may be enhanced by lower lithostatic pressures in the upper crust, where clays are important constituents of fault gouges, as the size and mobility of ripplocations are likely a function of confinement[21].

The existence of ripplocations has fundamental implications for our understanding of phyllosilicate deformation. Existing models of KB formation struggle to explain $c$-axis parallel strain without invoking brittle mechanisms. KBBs display a range of rotation angles and radii of curvature, and may be partially elastic, as shown by reversible experimental hysteresis curves[40]. Such 'incipient' KBs appear to be entirely reversible, up to a point where they are pinned and transition into permanent KBs. This process of recoverable plasticity is better explained by the formation and destruction of ripplocations than by the motion of basal dislocations or dislocation walls. The ability of ripplocations to store extensive elastic strain energy raises questions on how this strain may be distributed to other phases or released under changing conditions in the Earth. It is worth noting that while studies of natural rocks represent a 'frozen in' microstructure in which the contribution, or impact, of elastic

strain is difficult to quantify post-mortem, elastic strains are likely to play an important role in the development of locally high stresses with a significant effect on recrystallisation and mineral reactions[41,42]. It remains unclear whether ripplocations should be defined as either brittle or viscous defects (see also Supplementary Note 3). They are defined as waves in atomic layers[18–20] and can form at different scales, involving few to several interlayer bonds. Ripplocations may stretch interlayer bonds (visco-elastic behaviour), but larger-scale ripplocations may cause these bonds to break (delaminations as lenticular micro-cracks). If lenticular micro-cracks form, the motion of ripplocations might drive micro-crack migration in a continuous cycle of cracking and healing, thus accommodating deformation via a combination of visco-elastic and transient brittle processes. The behaviour we describe here would have considerable implications for fluid transport in deep shear zones where ripplocations would provide the transient porosity waves needed to move fluids through the crust. This idea is not dissimilar to the model proposed by Phipps Morgan and Holtzman[43] where vug waves are defined as a combined deformation and fluid migration mechanism, whereby migration of fluid-filled cracks is driven by the release of elastic strain energy. Fluids in faults, shear zones and subduction zones can weaken the mineral components of these structures, increase the rate of diffusive mass transfer, or generate overpressures. While the applicability of a vug wave-type model to ripplocations must be tested carefully with further work, the implications for the strength of faults, shear zones and subduction zones are significant. As the properties of ripplocation defects are clearly important for understanding the rheology of micas in mylonites of the middle and lower crust, so they may also prove a critical consideration for the frictional behaviour of phyllosilicates within seismogenic faults and subduction zones[44,45]. Now that ripplocations have been identified in natural phyllosilicates, these defects must be accounted for in microphysical modelling of the mechanical properties of phyllosilicate-rich faults, shear zones and subduction zones as they are likely to have a crucial influence on the rheology of these tectonic lineaments and on the nucleation and propagation of large-magnitude earthquakes[5,6,44–46].

## Methods

**Preparation of TEM samples.** Electron transparent TEM specimens were produced by both FIB and Ar ion milling in order to identify and compare any effects of sample preparation procedure on the distribution and morphology of delaminations. This comparison is important as FIB is a popular and effective method of TEM sample preparation, however the majority of previous studies on deformed phyllosilicates utilised Ar ion milling techniques. The FIB specimens were produced using the lift out method using a FEI Helios Nanolab™ 600 I DualBeam™ Ga FIB scanning electron microscope. Target biotite grains were identified in standard petrographic thin sections of mylonitic samples using optical and scanning electron microscopy. Site-specific specimens were prepared using the FIB, with sections cut perpendicular to cleavage via the in situ trenching and lift out technique[25] (Supplementary Fig. 7a). Once prepared, thin lamellae received a 5.00 kV low-energy surface wipe, to reduce the thickness of amorphised surface damaged layers. During this thinning process, all samples were observed to bend and flex extensively, which we interpret to result from the release of stored elastic strain energy (Supplementary Fig. 7b, c).

Samples prepared using Ar ion milling were taken from 3 mm cores drilled into hand specimens. These cores were then encased in brass tubes using crystalbond adhesive and sliced into ~0.5 mm-thick discs with a low-powered diamond bladed saw. The discs were then reduced to a thickness of ~100 μm by polishing with 600 and 1200 grit papers on a water-based automatic polisher and finished to a high quality using 2500 and 4000 grit papers on the same machine. The polished 100 μm-thick discs were removed from their protective brass rings and cleaned of crystalbond by immersion in acetone for several hours. Finally, selected discs were placed in a Gatan PIPS and milled with dual Ar ion beams until a perforation was formed within or bordering a biotite grain. Heterogeneous milling rates, due to the multiphase nature of the samples, complicated this process, with the first perforation sometimes not occurring in a biotite grain, meaning that further milling was required.

**TEM analysis.** Transmission electron microscopy experiments were performed using a CEOS GmbH 'CESCOR' probe side aberration-corrected JEOL 2100FCs instrument, operating at 200 keV. The resolution offered by the microscope is 1.4 Å using conventional TEM illumination and sub-Ångstrom using aberration-corrected scanning TEM.

Low-resolution imaging and EDX measurements were performed using a tungsten filament JEOL 2000FX TEM operating at 200 keV. Extra care was taken during TEM analysis due to the nature of micas as beam-sensitive materials and the fact that the delamination features of interest were initially hypothesised to be a result of beam damage. Beam exposure was kept to a minimum in areas of interest; alignments and focusing were carried out in sacrificial areas of the sample and critical regions exposed only during image capture. Some areas were imaged at multiple magnifications meaning they were exposed to the beam for longer. In these instances, beam damage was often observed but as described in the main text, this took the form of amorphisation and a loss of contrast. The expansion structures on which this study is focused were not observed to grow or form during extended exposure to the beam except in some regions in close proximity to the perforation edge of PIPS prepared samples.

Chemical analysis including beam-induced K loss measurements were performed using an EDAX EDX detector attached to the JEOL 2000FX TEM. Previously unexamined regions of biotite were selected for measurements to ensure no prior diffusion of K from beam exposure. Spectra were taken from target regions using collection times of either 90 live seconds or 120 live seconds, an amp time constant of 102.4 μs, detector area of 30 mm$^2$ and a detector energy resolution of 136 eV at full width at half maximum Mn K-alpha. Measurements were repeated at the same location using a constant live time to investigate the loss of K between each spectrum. During this period the beam was not moved from its position to ensure all spectra were collected from the same spot. Spectra were mathematically filtered to remove background data. K/Si ratios were used as described by Van der Pluijm et al.[35] so as to normalise the loss of K to that of Si.

## Data availability

The data associated with this study are available from the corresponding author upon request.

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

## Acknowledgements

This work was funded by a NERC studentship NE/L002469/1 with the Earth, Atmosphere and Oceans (EAO) doctoral training programme at the Universities of Liverpool. J.A. and E.M. would like to give special thanks to A. Kronenberg for insightful discussions and for bringing the work of M.W.B. and co-authors to their attention. A. Biggin, A. Tagliabue, R. Chiverrell and J. Mecklenburgh are thanked for critical reading of early versions of the manuscript. M.W.B. was funded by CMMI division of the NSF under grant 1728041.

## Author contributions

J.A. collected samples, prepared specimens, produced and analysed TEM data and wrote the manuscript. E.M. supervised the work, collected samples, provided guidance in image interpretation, data analysis and geological implications and helped write the manuscript. K.D. prepared FIB samples, produced TEM data and provided instruction and guidance on TEM analysis and interpretation. M.W.B. provided guidance on ripplocation theory and interpretation of microstructures. All contributed to improvements of the manuscript.

## Additional information

**Competing interests:** The authors declare no competing interests.

**Journal Peer Review Information** *Nature Communications* thanks Andre Niemeijer and Yongsheng Zhou for their contribution to the peer review of this work. Peer reviewer reports are available.

