## [Peer Review File · Nature Communications]

Reviewers' comments:

Reviewer #1 (Remarks to the Author):

Review of "Ripplocations as a new mechanism for the deformation of phyllosilicates in the lithosphere"

by Joe Aslin, Elisabetta Mariani, Karl Dawson and Michel W. Barsoum

This manuscript describes a series of observations of peculiar crystal defect in biotite crystals, called "ripplocations". The authors interpret these ripplocations to be the missing element needed to understand the deformation of phyllosilicates. The observations presented are clear and obvious care has been taken in an attempt to eliminate possible alternative explanations for the observations. I have the following questions, comments and concerns.

1) The title is somewhat misleading and leads to a bit of a mixed message in the manuscript. Ripplocations in and of themselves are not a deformation mechanism, but rather a new type of defect. It would be more correct to say that the movement of ripplocations is a new deformation mechanism in phyllosilicates. But even this isn't entirely correct, I feel. The authors write as if dislocation glide in the (001) plane in phyllosilicates is not possible without ripplocations, but if the phyllosilicates are perfectly aligned with the deformation direction in simple shear, I don't see a need for an additional mechanism. It's basically like sliding a deck of cards, which also makes you wonder what the contribution is purely frictional sliding is in the deformation of well-aligned phyllosilicates. As far as I can tell, the movement of ripplocations provides a mechanism for the formation of kinks and kink bands which allows out-of-plane deformation of the phyllosilicates. I feel that this should be made more clear in the title and throughout the text (in particular in the abstract and the sentence "dislocation creep in not a viable deformation mechanism in phyllosilicates").

I think that the main message is that the presence and movement of ripplocations might provide an explanation for kinking, but this message is not that different from what was concluded in the Gruber et al (Na Scientific Reports, 2016) paper. Still, the observation of these defects in geological materials is interesting in and of itself.

2) The key implication of the presence of ripplocations is that it allows for strain perpendicular to the basal plane to occur. The authors speculate that this might be the source of the dependence of yield strength on confining pressure, which is an interesting notion. However, there is another mechanism for the accommodation of normal strain which the authors did not consider. The operation of simple granular flow, that is grains rotating and sliding past each other can also generate volumetric strain and would also lead to a confining pressure dependent strength. It's of course entirely possible that the two mechanisms operate simultaneously, but I wonder if it's possible to provide an estimate of the degree of confining pressure would be expected. Presumably, the normal strains associated with ripplocations is much smaller than that associated with grain rotation and translation. Perhaps some strain energy considerations can be used? Additionally, the granular flow mechanism would probably depend on grain size, is there any data available on this?

3) I was surprised that the ripplocations were found in samples of westerly granite. If I understand the formation mechanism properly, a compressive stress would be needed to form these defects which is not present in the Westerly granite, which is pretty much undeformed. The authors write that ripplocations are a type of crystal defect and might therefore develop during crystal growth, but that seems at odds with the simulations of Gruber et al (2016) in which a compressive stress is needed to generate them.

4) The notion that ripplocations might play a role in the enhancement of pressure solution in the presence of phyllosilicate is pretty far-fetched in my opinion. These are very small-scale features so the associated potentially heterogenous stress will be small. It is pretty much established that the presence of phyllosilicates instead allow a thicker fluid film or layer to be present which leads to faster diffusion rates. If heterogenous stresses play a role, I would expect to get more heterogeneity from just the roughness of the quartz surfaces.

Reviewer #2 (Remarks to the Author):

The authors submitted a research article with a title " Rippllocations as a new mechanism for the deformation of phyllosilicates in the lithosphere".

This manuscript reports that nano-scale bending characteristic of rippllocations, as a new deformation mechanism, was found in phyllosilicates used high-resolution transmission electron microscopy (TEM), which was explained by dislocation glide in all previous studies.

What report in the manuscript is new found, and this topic is more attractive to readers. The manuscript is well prepared. So I suggest the manuscript suitable to publish in nature communication.

However, due to the following general comments, I would like to recommend it to be minor revision.

General comments:

1. What is the difference between rippllocation and dislocation under TEM?

The most important evidence in this study is nano-scale bending characteristic under high-resolution transmission electron microscopy (TEM). However, dislocation glide also can produce similar deformation structure in previous studies.

Authors argued that "Nano-scale expansion structures have the appearance of waves in Fig. 2c, as predicted by existing models of rippllocations and show characteristic bending of lattice fringes on the order of 30° within as little as 3 nm", and "Such sharp bending of layers is orders of magnitude greater than that which could be produced by individual dislocations or complex arrays of dislocation walls". Which means that sharp bending of layers can be found in both dislocation and rippllocation, but only difference in orders of magnitude of bending.

Another evidence for rippllocation is the geometric arrangement of delaminations. Authors argue that the best explanation is an overall expansion parallel to the c-axis (perpendicular to the basal plane) of the biotite grain, and basal dislocations cannot impart c-axis strain proposed rippllocations in this study as the mechanism governing the observed lattice bending. Is it correct? I am not sure. In my mind, dislocation glide process should be accompanied by expansion parallel to the c-axis.

2. The edges of 'diamond-shaped arrays' geometries was found only in this study and never report in previous studies. So, this kind of 'diamond -shaped arrays' was explained that stored elastic strain energy is released during thinning for TEM, not necessarily the time during which strain was being accumulated in deformation process within the viscous shear zone.

However, diamond-shaped arrays is mechanism to form kink band in mica, why was these 'diamond-shaped arrays' geometries produced during thinning for TEM?

Compared with deformation of mica in three deformed samples, 'diamond-shaped arrays' was not found in the mica from undeformed westerly granite, which also used the same method to prepared TEM samples, why rippllocation in sample of westerly granite did not occur during thinning for TEM

3. What is the deformation conditions of rippllocation? In ductile shear deformation or brittle deformation?

Authors suggested that the nucleation and motion of rippllocations is strongly dependent on the lithostatic pressure, which is mechanically brittle behavior. Fractures did not found but rather produce a bending of lattice planes and kinking, with the development of associated layer-normal strains. Therefore, in strongly aligned phyllosilicate rocks, the pressure sensitive strength of deforming phyllosilicates can be explained by rippllocations.

The edges of these 'diamond' geometries are 30° - 60° apart, it is clear comparison to conjugate normal faulting, which occurs under extensional stresses under brittle conditions.

However, the three naturally deformed samples used in this study were collected from mylonites which is viscously deformed granitic rocks, and it was active at lower amphibolite facies conditions. That means the mica in mylonites deformed under plastic conditions, which should be related to dislocation. This is conflict with the pressure sensitive brittle deformation.

Special comments:

"the nucleation and motion of ripplocations is strongly dependant on their confinement (or the lithostatic pressure)". Here, "dependant" should be dependent.

Response to Reviewers' comments

Pls note our comments are in blue while reviewers' points are in black *italics*.

Reviewer #1 (Remarks to the Author):

Review of "Ripplocations as a new mechanism for the deformation of phyllosilicates in the lithosphere"

by Joe Aslin, Elisabetta Mariani, Karl Dawson and Michel W. Barsoum

This manuscript describes a series of observations of peculiar crystal defect in biotite crystals, called "ripplocations". The authors interpret these ripplocations to be the missing element needed to understand the deformation of phyllosilicates. The observations presented are clear and obvious care has been taken in an attempt to eliminate possible alternative explanations for the observations. I have the following questions, comments and concerns.

We appreciate and welcome the Reviewer's comment recognising the care we have taken in our study, the rigour and attention to detail in both our systematic TEM observations and our interpretations. We did consider carefully all possible alternative explanations and favoured the interpretation that is best supported by our new experimental evidence. Below we address each individual point raised by the reviewer.

1a) *The title is somewhat misleading and leads to a bit of a mixed message in the manuscript. Ripplocations in and of themselves are not a deformation mechanism, but rather a new type of defect. It would be more correct to say that the movement of ripplocations is a new deformation mechanism in phyllosilicates. But even this isn't entirely correct, I feel.*

We agree with the reviewer's assertion that an individual ripplocation is a defect rather than a mechanism itself and we state this clearly in the manuscript (See line 13 in the Abstract, and the section on "Ripplocations in layered materials"). The mechanism we refer to is the motion of ripplocations and their interactions, both of which facilitate shape-change and previously unexplained c-axis parallel strain in phyllosilicates.

Following the reviewer's comment we have changed the manuscript title to: "Ripplocations provide a new mechanism for phyllosilicate deformation in the lithosphere". We have also changed: "represent" to "provide" (line 78), "Ripplocations and their motion" to "Ripplocation motion and their interactions" (line 83) and "mechanism" to "defects" (line 193).

We would also like to highlight the following important points:

- as ripplocations are new defects, there is currently no accepted term for the mechanism of ripplocation motion (equivalent to dislocation creep for example);
- In this study we do not observe, nor do we measure the motion of ripplocations but we do observe for the first time the effects of this mechanism in mylonitic (i.e. highly deformed) biotite and the micro- (and nano-) structures associated with it.

1b) *The authors write as if dislocation glide in the (001) plane in phyllosilicates is not possible without ripplocations, but if the phyllosilicates are perfectly aligned with the deformation direction in simple shear, I don't see a need for an additional mechanism. It's basically like sliding a deck of cards, which*

also makes you wonder what the contribution is purely frictional sliding is in the deformation of well-aligned phyllosilicates.

While the concept of dislocation glide along the basal plane (simply “like sliding on a deck of cards” as the reviewer mentions) of phyllosilicate minerals has been discussed by the scientific community since the 1970s or earlier, its viability in such complex mineral structures has been questioned and the need for alternative mechanisms has been voiced for several decades (e.g. Etheridge et al. 1973). Importantly, we would like to bring to the Editor’s and Reviewer’s attention that there is a common but unjustified assumption in the literature that dislocation glide in micas occurs along the interlayer (e.g. Bell et al. 1986), which is also the cleavage plane in these minerals. Noe & Veblen (1999) present a detailed study, based on energy balance considerations, that shows that dislocations are *not possible* in the cation interlayer of biotite and that they are most energetically viable in the oxygen layer between octahedral and tetrahedral sheets. The study of Noe and Veblen highlights the fact that the idea of dislocations as introduced by materials scientists in the 1930s does not apply simply to layered silicate minerals. This is because in these minerals simple, highly symmetrical half atomic planes are replaced by complex arrangements of Si-Al tetrahedral and octahedral structures, forming layers, and linear defects in these structures may look like ripples rather than dislocations as we know them. There is therefore a strong motivation to reconsider how dislocations can function at all in complex structures such as those of layered minerals and layered materials in general. Figure 1 below shows diagrams of ripplocations in biotite and illustrates how they might compare with dislocations.

Figure 1.

In this study we show that ripplocations are active in the (001) interlayer plane and, based on the findings of Noe & Veblen (1999), we propose that they can facilitate deformation without any need for dislocations or dislocation glide in this plane. We do not argue, however, against dislocation glide in the (001) oxygen layer because this was not the primary goal of our study and we did not gather evidence for or against it as this would require new, time intensive and challenging experiments. For

ease of writing, throughout the manuscript we have chosen to refer to the glide plane simply as the (001) or basal plane.

We propose that the formation and motion of ripplocations enable deformation both parallel and perpendicular to the basal plane. This might not preclude basal dislocation glide to occur simultaneously to ripplocation motion. However, dislocation glide on its own (just like in a deck of cards) must be a short-lived mechanism, its demise caused by the presence of grain boundaries, obstacles and imperfect alignment of mica grains in nature. We expect a similar fate would happen to frictional sliding mechanisms that also require the perfect alignment of micas.

Ripplocation models (e.g. Gruber et al. 2016) show that, in a single crystal, these defects behave like waves and can form, merge, migrate and annihilate in a layered structure. However the behaviour of ripplocations across grain boundaries is yet to be investigated.

Based on our observations of micro- and nano-structures in a polycrystalline mica-rich mylonite, we conclude that ripplocations must be ubiquitous in phyllosilicates. To clarify the role of dislocations and dislocation glide we have added lines 37-40 and lines 44-49.

In the figure below we further differentiate between ripplocations and dislocations. Let us start by making a cut – normal to the basal planes - in a layered crystal. When an extra plane is added here, ordinarily it would produce an edge dislocation, however, as this crystal contains weakly bonded interlayers it is reasonable to assume that the “extra material” will cause a ripple in the basal planes shown in Fig. 1 and exaggeratedly in lower left corner of Fig. 2 below. This would then be a ripple boundary rather than a dislocation but there may be no reason to define a “new” defect. Where this logic breaks down however, is when a second “ripple boundary” or “dislocation” of the same sign is added (as in the lower right of Fig. 2): ripplocations have been shown –by Kushima et al. (2015) and Gruber et al. (2016) - **to attract and merge**. On the contrary, according to elementary dislocation theory, two basal dislocations of the same sign would repel. These “ripple boundaries” are thus fundamentally different from dislocations. The complex nature of micas means that this does not preclude the presence of complicated screw dislocation networks in the basal planes, however as discussed in our study, these cannot account for the bending and/or c-axis strain we observed.

Figure 2: Why dislocations cannot exist in the interlayer plane of biotite and in the structure of other layered solids.

1c) As far as I can tell, the movement of ripplocations provides a mechanism for the formation of kinks and kink bands which allows out-of-plane deformation of the phyllosilicates. I feel that this should be made more clear in the title and throughout the text (in particular in the abstract and the sentence “dislocation creep is not a viable deformation mechanism in phyllosilicates”). I think that the main message is that the presence and movement of ripplocations might provide an explanation for kinking, but this message is not that different from what was concluded in the Gruber et al (Na Scientific Reports, 2016) paper. Still, the observation of these defects in geological materials is interesting in and of itself.

We appreciate the reviewer’s comment that the recognition of ripplocation defects in geological materials is a discovery of interest in its own right. We would also like to emphasise that the message of our study, which demonstrates the existence of ripplocations in phyllosilicates for the first time, differs substantially from that of the Gruber et al. (Scientific Reports 2016) paper, which focusses on modelling the behaviour of ripplocations in synthetic materials.

We have clarified what we mean with “dislocation creep is not a viable deformation mechanism in phyllosilicates” by adding lines 37 to 40 on page 2 and also explained this in response to the Reviewer’s comment 1b above.

We agree with the reviewer that ripplocations provide an excellent means to explain the sharp bending during formation of kink band boundaries (KBBs) and kink bands (KBs). Our study is the first to explain the geometric compatibility problem in mica kink bands first highlighted by Etheridge et al. (1973). We do so using carefully planned high resolution TEM experiments. Gruber et al. (2016) also come to the conclusion that ripplocations can stack efficiently to form KBBs, but they do so using ripplocation modelling in synthetic materials. Furthermore, our study goes beyond the work of Gruber et al. (2016) because we investigate intensely sheared biotite, sourced from a viscous shear zone in the Italian Western Alps, which is understood to have accommodated several 10s of km of displacement (Rutter et al. 2007). The high density of ripplocations observed in this biotite strongly suggests that, as well as forming KBBs (that are thought to accommodate low strains), ripplocations must move and interact to provide a new deformation mechanism able to accommodate large strains in such tectonic shear zones. This statement is supported by the recognition that, if ripplocations can be modelled as waves (Gruber et al. 2016), then by definition they are mobile (see also Morgan & Holtzman 2005). Beyond a handful of studies, very little is known about ripplocation behaviour (mobility, interactions and transport of matter) and, as this is an exciting new field, we anticipate that our study will stimulate a flurry of activity to shed light on such processes.

We would like to point out to Reviewer 1 that out-of-plane deformation is inherent to ripplocations (see also Figure 1), even when they do not form KBs. Ripplocations can therefore explain the confining pressure sensitivity of micas (as discussed below) and may have implications for the storage of elastic strain energy in phyllosilicate bearing faults and shear zones.

We feel we have highlighted the importance of ripplocations for KB formation sufficiently in the introductory section in lines 50-56, lines 102-113 and lines 242-248 as well as in Figure 4 and in Supplementary Information on “Kink band morphology and formation”. We also feel that strengthening this aspect any further would be limiting because, as we have explained, ripplocations do have considerable broader importance.

We have clarified the broader importance of ripplocations, beyond KB formation, in additional text in lines 214-223 and lines 254-270.

For all reasons above we have not changed the title to accommodate the KB formation process.

2) The key implication of the presence of ripplocations is that it allows for strain perpendicular to the basal plane to occur. The authors speculate that this might be the source of the dependence of yield strength on confining pressure, which is an interesting notion. However, there is another mechanism for the accommodation of normal strain which the authors did not consider. The operation of simple granular flow, that is grains rotating and sliding past each other can also generate volumetric strain and would also lead to a confining pressure dependent strength. It's of course entirely possible that the two mechanisms operate simultaneously, but I wonder if it's possible to provide an estimate of the degree of confining pressure would be expected. Presumably, the normal strains associated with ripplocations is much smaller than that associated with grain rotation and translation. Perhaps some strain energy considerations can be used? Additionally, the granular flow mechanism would probably depend on grain size, is there any data available on this?

The reviewer makes an interesting point here. We agree that granular flow can generate volumetric strain, and it is therefore a brittle mechanism that could account for the pressure sensitivity of mica-bearing rocks (e.g. Figure 3 below). However, it is important in this respect to note that the confining pressure dependent strength is observed at the single crystal level (e.g. see Christofferson and Kronenberg, 1993) as well as in polycrystalline aggregates (e.g. Mariani et al., 2006). Granular flow cannot account for intracrystalline c-axis parallel strain or, therefore, the pressure dependency of single crystals. The process of granular flow (or cataclastic flow) involves a significant amount of grain comminution and fracturing, producing a characteristic microstructure, marked for example by the disaggregation of quartz or feldspar grains in fault rocks (See Fig. 3 below).

Figure 3. Granular flow in the graphitic schists exposed along the Carboneras Fault Zone, Almeria, Spain. Photo by E. Mariani, 2012.

We have not seen such granular flow microstructure in deformed highly micaceous layers, which instead respond to deformation by intense, high angle kinking as shown in Fig. 1 c of Supplementary Information. These comments notwithstanding, there is no reason why granular flow cannot occur together with ripplocation motion. Actually one can make the argument that one triggers the other and vice versa.

While we do not think it is possible to include strain energy considerations in the current manuscript (because this would amount to a new study), we would welcome a collaboration with Reviewer 1,

should they be interested, on determining the strain energy associated with ripplocations and related kinks versus that of granular flow, in phyllosilicate-rich rocks where these processes may be observed to occur in series or in parallel.

We have added lines 214-223 to clarify further the contribution of ripplocations to a pressure-sensitive response to deformation.

We are somewhat confused by the Reviewer's comment: "the granular flow mechanism would probably depend on grain size". The reviewer might be referring here to another "type" of granular flow, which pertains to viscous deformation, whereby grains rotate and translate relative to each other through grain boundary sliding (Paterson 1995; Paterson 2001; Rutter 1997). As this is a viscous deformation process it does not involve dilation, meaning it is not a pressure-sensitive process. Therefore, it cannot explain the pressure sensitivity of phyllosilicates.

3) *I was surprised that the ripplocations were found in samples of westerly granite. If I understand the formation mechanism properly, a compressive stress would be needed to form these defects which is not present in the Westerly granite, which is pretty much undeformed. The authors write that ripplocations are a type of crystal defect and might therefore develop during crystal growth, but that seems at odds with the simulations of Gruber et al (2016) in which a compressive stress is needed to generate them.*

In response to this comment by Reviewer 1 we have added lines 132-143 to clarify how ripplocations may form in biotite in a granite.

It must be noted that there is well-established theory that states dislocation defects themselves form under stress (like ripplocations) from Frank-Reed sources (e.g. Hirth & Lothe 1982). This however does not preclude that dislocations could also form as growth defects that are geometrically necessary to accommodate a lattice mismatch in imperfect crystals (Matthews & Blakeslee 1974).

Similarly, while ripplocations do develop under stress in the simulations of Gruber et al. (2016) this does not preclude that, like dislocations, they could also form to accommodate a layer mismatch during growth of layered materials, for example in a granitic melt, or when localised differential stresses are produced during cooling or uplift.

As an additional observation, the lack of kinked biotite grains in Westerly granite suggests that, while ripplocations are present in this biotite, no motion or interaction of ripplocations have occurred. This may support the idea that here ripplocations are growth defects rather than caused by deformation. For clarity we have added this observation between lines 139-141 in the manuscript. We have also re-made Supplementary Figure 5 to show more explicitly the presence of delaminations in the Westerly granite sample.

4) *The notion that ripplocations might play a role in the enhancement of pressure solution in the presence of phyllosilicate is pretty far-fetched in my opinion. These are very small-scale features so the associated potentially heterogenous stress will be small. It is pretty much established that the presence of phyllosilicates instead allow a thicker fluid film or layer to be present which leads to faster diffusion rates. If heterogenous stresses play a role, I would expect to get more heterogeneity from just the roughness of the quartz surfaces.*

This is a valid comment and we do recognise that, until the elastic strain energy stored in ripplocations can be measured (or estimated), there is no supporting evidence to show that such strain energy is sufficiently large to play a role in facilitating pressure solution. In agreement with the Reviewer's comment we have removed the sentences: "Phyllosilicates are shown to enhance pressure solution processes in other minerals they come in contact with (Hickman & Evans 1995). The microphysical controls on this are unclear, but locally high stresses could be major contributors to facilitating pressure solution reactions (Wheeler 2014; Wheeler 2018)".

Reviewer #2 (Remarks to the Author):

The authors submitted a research article with a title " Ripplocations as a new mechanism for the deformation of phyllosilicates in the lithosphere".

This manuscript reports that nano-scale bending characteristic of ripplocations, as a new deformation mechanism, was found in phyllosilicates used high-resolution transmission electron microscopy (TEM), which was explained by dislocation glide in all previous studies.

What report in the manuscript is new found, and this topic is more attractive to readers. The manuscript is well prepared. So I suggest the manuscript suitable to publish in nature communication.

However, due to the following general comments, I would like to recommend it to be minor revision.

We greatly appreciate the reviewer's support for the publication of this manuscript and their recognition of the importance of these findings to readers of Nature Communications. We have addressed their constructive comments below.

General comments:

1a. *What is the difference between ripplocation and dislocation under TEM?*

The most important evidence in this study is nano-scale bending characteristic under high-resolution transmission electron microscopy (TEM). However, dislocation glide also can produce similar deformation structure in previous studies.

Authors argued that "Nano-scale expansion structures have the appearance of waves in Fig. 2c, as predicted by existing models of ripplocations and show characteristic bending of lattice fringes on the order of 30° within as little as 3 nm", and "Such sharp bending of layers is orders of magnitude greater than that which could be produced by individual dislocations or complex arrays of dislocation walls". Which means that sharp bending of layers can be found in both dislocation and ripplocation, but only difference in orders of magnitude of bending.

We refer the reviewer to Fig. 1 and 2 above concerning basal dislocations and why it would be unlikely that they exist in the interlayer plane of biotite and the basal planes of other layered solids. As noted above, ripplocations do not preclude the presence of complicated screw dislocation networks within the various basal planes. The presence of the latter however, cannot explain the c-axis strain observed, but they can certainly be mistaken for basal dislocations.

In previous studies dislocations have nominally been observed by *in-situ* TEM as curved line defects moving on mica basal planes (more precisely, on mica cleavage, or interlayer, planes – see Meike (1989), but also Mares & Kronenberg (1993), Christoffersen & Kronenberg (1993). Dislocations may also be visible by *atomic resolution* TEM on a plane orthogonal to layering, as shown in the work of

Noe & Veblen (1999). Rippllocations, unlike dislocations, can range in size from nm to μm scale (e.g. Figure 2 in the manuscript and Figure 1 above).

As a note we would like to point out that the work of Meike (1989) and Noe & Veblen (1999), while both excellent, come to incompatible conclusions. Meike claims to image dislocations gliding on the cleavage (interlayer) planes. Noe & Veblen conclude dislocations on such planes cannot be explained by energy calculations and therefore they are unlikely to exist there.

If the mobile line defects observed by Meike (and other authors) on interlayer planes were rippllocations (but these authors did not know they existed), this would reconcile their observations with those of Noe & Veblen (1999) and with our findings in this study.

In the schematic diagram of Figure 1 above we show how rippllocations could compare with dislocations at the atomic scale. It must be noted that this diagram is drawn with view orthogonal to layering and does not account for the additional complexities represented by the ring structures of tetrahedrons and octahedrons that form basal planes.

We would like to point out that there are *two* (not just one) important bodies of evidence in our manuscript for the existence of rippllocations: The crystallographically-controlled patterns of diamond shaped arrays in biotite grains *and* the nano-scale, high magnitude, lattice bending (identified by the reviewer).

Below we explain why, contrary to what is stated by the reviewer, dislocation glide *cannot* produce the nano-scale bending observed.

Our key argument, laid out in Supplementary Information (Note 1), is that the high magnitude of rotation we observed at the nano-scale (30° bending within as little as 3 nm) is evidence that only rippllocation defects can produce such bending, not dislocations. This statement is supported by the fact that, to achieve 30° bending around 15 dislocations are required, each accommodating a curvature of 2° in the basal plane (as indicated by e.g. Bell et al. 1986). *However, 15 dislocations cannot physically fit within 3 nm length of biotite lattice, corresponding to less than 8 biotite unit cells in the a direction, or 4 unit cells in the b direction.*

To make our argument clearer we have moved part of the content of Supplementary Note 1 to the main text (lines 102-113).

We would also like to thank the reviewer for highlighting a potential ambiguity in our use of the word “sharp”. The reviewer is correct that in the manuscript we do use “sharp” to indicate the magnitude of bending (i.e. a high rotation angle), not a smooth versus sharp curvature (which can be smooth or sharp for any magnitude of the bending angle). We have clarified this by changing text between lines 98-102.

Along these lines, Freiberg et al. (2018) defined reversible rippllocation boundaries, and made the case that they are the precursor of kink boundaries that are sharp and permanent. This natural progression of a rippllocation boundary to a sharper kink boundary is inherent to our ideas and cannot be explained using basal dislocations (see for example Supplementary Figures 1a and b).

1b. *Another evidence for rippllocation is the geometric arrangement of delaminations. Authors argue that the best explanation is an overall expansion parallel to the c-axis (perpendicular to the basal plane) of the biotite grain, and basal dislocations cannot impart c-axis strain proposed rippllocations*

in this study as the mechanism governing the observed lattice bending. Is it correct? I am not sure. In my mind, dislocation glide process should be accompanied by expansion parallel to the c-axis.

Dislocation glide itself cannot accommodate c-axis strain as the motion of dislocations is confined to the basal plane, there is no facility in basal dislocation glide for material to move parallel to the c-axis (see classic work of e.g. Frost & Ashby 1982; Poirier 1985). Therefore another mechanism is required. As the reviewer notes above, we argue that the geometric arrangement of arrays of expansion structures and delaminations suggests overall expansion of the c-axis. Basal dislocations cannot store the c-axis energy required to cause such expansion, whereas mobile ripplocations arranged throughout the biotite lattice could produce exactly this effect, as shown by the simulations of Gruber et al. (2016) and our TEM results.

2a. *The edges of 'diamond-shaped arrays' geometries was found only in this study and never report in previous studies. So, this kind of 'diamond-shaped arrays' was explained that stored elastic strain energy is released during thinning for TEM, not necessarily the time during which strain was being accumulated in deformation process within the viscous shear zone.*

However, diamond-shaped arrays is mechanism to form kink band in mica, why was these 'diamond-shaped arrays' geometries produced during thinning for TEM?

The delamination arrays *are not* the mechanism which forms kink bands. Rather, the mechanism we invoke for kinking is the motion of ripplocations lining up across multiple layers to form a through-going bend in the lattice under compressive stress. Let's now consider phyllosilicates where the lattice has been deformed sufficiently to form ripplocations but not for them to align into kink bands. In such layered minerals, on releasing the confining pressure during thinning for TEM analysis (i.e. removing more and more of the material that confines grains in a rock), the energy stored in the confined ripplocations is released in the form of grain-scale c-axis expansion, producing arrays of diamond shaped structures and, in some cases, delaminations, flanked by tight lattice curvature. Therefore, arrays of diamond shaped expansion structures and delaminations are merely the consequence of removing the forces that constrain the elastic strain energy stored in ripplocations. An analogy could be a spring constrained between 2 concrete blocks. If we remove the blocks the spring extends to its full length, releasing the elastic strain energy it was exerting against the blocks. This is explained in Fig. 4 in the manuscript.

2b. *Compared with deformation of mica in three deformed samples, 'diamond-shaped arrays' was not found in the mica from undeformed westerly granite, which also used the same method to prepared TEM samples, why ripplocation in sample of westerly granite did not occur during thinning for TEM*

We explain in lines 129-132 that expansion structures and delaminations were observed in the Westerly granite samples albeit in lower abundance than in the mylonitic samples, and that the diamond shaped arrays were present but less prominent in the Westerly granite samples (see also our reply 3 to Reviewer #1). We have adapted the wording here to make this clearer.

3. *What is the deformation conditions of ripplocation? In ductile shear deformation or brittle deformation?*

Authors suggested that the nucleation and motion of ripplocations is strongly dependent on the lithostatic pressure, which is mechanically brittle behavior. Fractures did not found but rather produce a bending of lattice planes and kinking, with the development of associated layer-normal

strains. Therefore, in strongly aligned phyllosilicate rocks, the pressure sensitive strength of deforming phyllosilicates can be explained by ripplocations.

The edges of these 'diamond' geometries are 30°-60° apart, it is clear comparison to conjugate normal faulting, which occurs under extensional stresses under brittle conditions.

However, the three naturally deformed samples used in this study were collected from mylonites which is viscously deformed granitic rocks, and it was active at lower amphibolite facies conditions.

That means the mica in mylonites deformed under plastic conditions, which should be related to dislocation. This is conflict with the pressure sensitive brittle deformation.

The reviewer here raises an excellent point: Are ripplocations brittle or viscous defects? We had started a discussion of this in Supplementary Note 3, but we do admit that our ideas are in an embryonic state. In this manuscript we are exploring such a novel field that textbook definitions of brittle and viscous deformation break down. We must therefore keep our mind free of any pre-conceived ideas in order to be able to interpret our observations objectively.

It is recognised in the literature that deformation in phyllosilicates is special, and different to the deformation behaviour of most other non-layered solids (e.g. Kronenberg et al. 1990; Mares & Kronenberg 1993; Mariani et al. 2006). In Phyllosilicates both viscous (dislocation glide) and brittle (cracks associated with kinking and dilation) deformation mechanisms are seen to occur through a large range of conditions in the Earth (from the Earth surface to the middle and lower crust, e.g. (Meike 1989; Kronenberg et al. 1990; Mares & Kronenberg 1993; Mariani et al. 2006). In addition to this, while pressure sensitivity might traditionally be associated with brittle deformation, it occurs in micas even when no obvious microstructural evidence for brittle deformation can be observed (Mariani et al. 2006). *Essentially, micas appear to deform by a pressure sensitive but non-brittle mechanism at a range of conditions.* This has to date been a poorly understood phenomenon. *Ripplocations resolve this conflict by providing a pressure sensitive mechanism of deformation that does not result in fracturing or other characteristic brittle microstructures.*

It remains however unclear whether ripplocations should be defined as either brittle or viscous defects. They are defined as waves in atomic layers and can form at different scales, involving few to several interlayer bonds. Ripplocations may stretch interlayer bonds (viscoelastic behaviour), but larger scale ripplocations may cause these bonds to break (brittle micro-cracking). If lenticular micro-cracks form, the motion of ripplocations might also cause these cracks to heal in a continuous cycle, thus accommodating deformation via a combination of viscoelastic and transient brittle processes. The behaviour we describe here would also have considerable implications for fluid transport in deep shear zones where ripplocations could provide the transient porosity waves needed to move fluids through the crust even under relatively high lithostatic pressures. It is clear that we need to understand more about ripplocation behaviour.

Special comments:

“the nucleation and motion of ripplocations is strongly dependant on their confinement (or the lithostatic pressure).”. Here, “dependant” should be dependent.

We thank the reviewer and have corrected the typo.

Any additional text highlighted in yellow in the revised manuscript represents:

- Previous Figure 2 has been split into a new Figure 2 and Figure 3 with the addition of Fig. 2 panel c. This is to produce simpler figures that are clearer and show the structures in sufficient detail.

- Changes were made to correct typos we found and improve grammar.
- Addition of Supplementary Figure 4. to further highlight the abundance and morphology of expansion structures.
- Supplementary Figure 5 is adapted to better show the extent of expansion structures in granitic biotite.
- Format changes were made to comply with the Nature Communications formatting checklist (sections now titled 'Results' and 'Discussion').

References:

- Bell, I.A. et al., 1986. Kinks in mica: Role of dislocations and (001) cleavage. *Tectonophysics*, 127(1–2), pp.49–65.
- Billia, M.A. et al., 2013. Grain boundary dissolution porosity in quartzofeldspathic ultramylonites: Implications for permeability enhancement and weakening of mid-crustal shear zones. *Journal of Structural Geology*, 53, pp.2–14. Available at: <http://dx.doi.org/10.1016/j.jsg.2013.05.004>.
- Christoffersen, R. & Kronenberg, A.K., 1993. Dislocation interactions in experimentally deformed biotite. *Journal of Structural Geology*, 15(9–10), pp.1077–1095.
- Etheridge, M.A., Hobbs, B.E. & Paterson, M.S., 1973. Experimental deformation of single crystals of biotite. *Contributions to Mineralogy and Petrology*, 38(1), pp.21–36.
- Freiberg, D., Barsoum, M.W. & Tucker, G.J., 2018. Nucleation of ripplocations through atomistic modeling of surface nanoindentation in graphite. *Physical Review Materials*, 2(5), p.053602. Available at: <https://journals.aps.org/prmaterials/pdf/10.1103/PhysRevMaterials.2.053602> <https://link.aps.org/doi/10.1103/PhysRevMaterials.2.053602>.
- Frost, H.J. & Ashby, M.F., 1982. *Deformation mechanism maps: the plasticity and creep of metals and ceramics.*, Pergamon press.
- Gruber, J. et al., 2016. Evidence for Bulk Ripplocations in Layered Solids. *Scientific Reports*, 6(August), pp.1–8.
- Hickman, H. & Evans, B., 1995. Kinetics of pressure solution at halite-silica interfaces and intergranular clay films. *Journal of Geophysical Research*, 100(13), pp.113–132.
- Hirth, J.P. & Lothe, J., 1982. *Theory of Dislocations*, New York: John Wiley & Sons.
- Kronenberg, A.K., Kirby, S.H. & Pinkston, J., 1990. Basal slip and mechanical anisotropy of biotite. *Journal of Geophysical Research*, 95(B12), p.19257.
- Mares, V.. & Kronenberg, a. ., 1993. Experimental deformation of muscovite. *Journal of Structural Geology*, 15(9–10), pp.1061–1075.
- Mariani, E., Brodie, K.H. & Rutter, E.H., 2006. Experimental deformation of muscovite shear zones at high temperatures under hydrothermal conditions and the strength of phyllosilicate-bearing faults in nature. *Journal of Structural Geology*, 28, pp.1569–1587.
- Matthews, J.W. & Blakeslee, A.E., 1974. Defects in epitaxial multilayers I. Misfit dislocations. *Journal of Crystal Growth*, 27, pp.118–125.
- Meike, A., 1989. In situ deformation of micas: a high-voltage electron-microscope study. *American Mineralogist*, 74(7–8), pp.780–796.

- Morgan, J.P. & Holtzman, B.K., 2005. Vug waves: A mechanism for coupled rock deformation and fluid migration. *Geochemistry, Geophysics, Geosystems*, 6(8).
- Noe, D.C. & Veblen, D.R., 1999. HRTEM analysis of dislocation cores and stacking faults in naturally deformed biotite crystals. *American Mineralogist*, 84(11–12), pp.1925–1931.
- Paterson, M.S., 2001. A granular flow theory for the deformation of partially molten rock. *Tectonophysics*, 335, pp.51–61.
- Paterson, M.S., 1995. A theory for granular flow accommodated by material transfer via an intergranular fluid. *Tectonophysics*, 245, pp.135–151.
- Poirier, J.-P., 1985. *Creep of crystals: high-temperature deformation processes in metals, ceramics and minerals*, Cambridge University Press.
- Rutter, E. et al., 2007. Large-scale folding in the upper part of the Ivrea-Verbano zone, NW Italy. *Journal of Structural Geology*, 29(1), pp.1–17. Available at: <http://www.sciencedirect.com/science/article/pii/S0191814106002100>.
- Rutter, E.H., 1997. The influence of deformation on the extraction of crustal melts: a consideration of the role of melt-assisted granular flow. In *Deformation-enhanced Fluid Transport in the Earth's Crust and Mantle*. Mineralogical Society Series 8, pp. 82–110.
- Wheeler, J., 2014. Dramatic effects of stress on metamorphic reactions. *Geology*, 42(8), pp.647–650. Available at: <http://geology.gsapubs.org/cgi/doi/10.1130/G35718.1>.
- Wheeler, J., 2018. The effects of stress on reactions in the Earth: Sometimes rather mean, usually normal, always important. *Journal of Metamorphic Geology*, 36(4), pp.439–461.

REVIEWERS' COMMENTS:

Reviewer #1 (Remarks to the Author):

The authors have provided a very clear and complete reply to the concerns raised by me and the other reviewer. The answers are satisfactory and I have no further remarks, other than to complement the authors on an original and interesting piece of work.

Andre Niemeijer

Reviewer #2 (Remarks to the Author):

The authors resubmitted revised manuscript with new title "Ripplocations provide a new mechanism for the deformation of phyllosilicates in the lithosphere". This revised manuscript is well prepared, and made changes both for the text and Supplementary information based on most of comments by reviewers. The authors replied all of the questions in "Response to Reviewers' comments". I suggest the manuscript is good enough to publish in nature communication. So I think it should be accepted now.

Yongsheng Zhou

Response to Reviewers' comments

REVIEWERS' COMMENTS:

Reviewer #1 (Remarks to the Author):

The authors have provided a very clear and complete reply to the concerns raised by me and the other reviewer. The answers are satisfactory and I have no further remarks, other than to complement the authors on an original and interesting piece of work.

Andre Niemeijer

Reviewer #2 (Remarks to the Author):

The authors resubmitted revised manuscript with new title "Ripplocations provide a new mechanism for the deformation of phyllosilicates in the lithosphere".

This revised manuscript is well prepared, and made changes both for the text and Supplementary information based on most of comments by reviewers. The authors replied all of the questions in "Response to Reviewers' comments". I suggest the manuscript is good enough to publish in nature communication.

So I think it should be accepted now.

Yongsheng Zhou

We would very much like to thank both Dr Andre Niemeijer and Dr Yongsheng Zhou for their consideration of our work and for their constructive comments which have helped to improve our finalised manuscript. We are very pleased to hear that they were satisfied with our responses to their previous concerns and have no further issues to raise.

Joe Aslin, (on behalf of the Authors)